# Physical reality of the Preisach model for organic ferroelectrics

Indrė Urbanavičiūtė [1], Tim D. Cornelissen [1], Xiao Meng[2], Rint P. Sijbesma [2] & Martijn Kemerink [1]

The Preisach model has been a cornerstone in the fields of ferromagnetism and ferroelectricity since its inception. It describes a real, non-ideal, ferroic material as the sum of a distribution of ideal 'hysterons'. However, the physical reality of the model in ferroelectrics has been hard to establish. Here, we experimentally determine the Preisach (hysteron) distribution for two ferroelectric systems and show how its broadening directly relates to the materials' morphology. We connect the Preisach distribution to measured microscopic switching kinetics that underlay the macroscopic dispersive switching kinetics as commonly observed for practical ferroelectrics. The presented results reveal that the in principle mathematical construct of the Preisach model has a strong physical basis and is a powerful tool to explain polarization switching at all time scales in different types of ferroelectrics. These insights lead to guidelines for further advancement of the ferroelectric materials both for conventional and multi-bit data storage applications.

[1] Complex Materials and Devices, Department of Physics, Chemistry and Biology (IFM), Linköping University, SE-58 183 Linköping, Sweden. [2] Laboratory of Macromolecular and Organic Chemistry, Eindhoven University of Technology, P.O. Box 513, 5600 MB Eindhoven, The Netherlands. These authors contributed equally: Indrė Urbanavičiūtė, Tim D. Cornelissen. Correspondence and requests for materials should be addressed to M.K. (email: martijn.kemerink@liu.se)

Although originally developed to describe magnetic hysteresis, the Preisach model[1] has also successfully been applied to describe hysteretic switching in ferroelectrics[2–4]. The basic assumption of the Preisach model is that a macroscopic ferroelectric consists of a distribution of microscopic hysterons with a rectangular hysteresis loop. Each hysteron is characterized by its up-switch field $U$ and down-switch field $V$, defining its position on the corresponding Preisach plane as shown in Fig. 1a. While the model is mathematically transparent and easy to implement, the physical reality of the constituent hysterons has been hard to establish[2,5].

Most research on the Preisach model for ferroelectrics has been on inorganics, and focused on macroscopic[2,3] or local measurements[6,7] of the hysteron distribution, as well as on using the model to predict hysteresis loop behavior[4,8–10]. There are few applications of the model on organic ferroelectrics[11–15], and these have been limited to the polymer PVDF and its copolymers[10,16–19]. However, the physical reasons behind the shape and width of the distribution and especially the implications of the broadening of the Preisach distribution (PD) for other ferroelectric properties such as the switching kinetics are rarely discussed.

Switching kinetics in ferroelectrics are generally described by nucleation and growth mechanisms. It has been repeatedly reported that in the case of polycrystalline (disordered) ferroelectrics, the unrestricted domain growth (as assumed by the classical Kolmogorov–Avrami–Ishibashi (KAI) theory[20,21]) is hardly attainable[22–25]. This results in polarization switching current transients that are broader than predicted by the theory. To be able to successfully reproduce the switching kinetics, a distribution in characteristic switching times must be taken into account. While different types of these distributions are found to fit the experimental data, the physical reasoning behind them remains complex and requires several assumptions[23–28].

Here, by probing two ferroelectric materials with an entirely distinct structure, we show that the shape of the PD is directly related to the materials' nanostructure. As the diversity of nanostructure is relatively narrow among inorganic ferroelectric materials, which are mainly (poly)crystalline, we choose organic analogues—the semi-crystalline copolymer P(VDF-TrFE) and the polycrystalline molecular ferroelectric trialkylbenzene-1,3,5-tricarboxamide (BTA)[25,29,30]. We find that for BTA, which consists of close-packed dipolar columns (Fig. 2a), the PD is circular, which contrasts with a narrow ellipsoid PD of the copolymer P(VDF-TrFE), which comprises crystalline dipolar domains separated by amorphous regions (Fig. 2b). Since these materials are used as different model systems rather than as specific compounds, this increases the general relevance of our findings. The measured different PD shapes are quantitatively explained by a 3D dipole interaction model. The distribution parameters are further interpreted from the perspective of the structural and energetic disorder. We furthermore investigate the switching kinetics of discrete parts of the PD and find that the characteristic switching time is strongly dependent on the Preisach coordinates for both BTA and P(VDF-TrFE). We prove that this variation is the origin of the broad switching time distribution that underlies the dispersive switching kinetics. Based on the experimental observations, we show that the combination of the Preisach model, the thermally-activated nucleation-limited switching (TA-NLS) formalism, and the adapted Kolmogorov–Avrami–Ishibashi (KAI-NLS) theory provides a full and consistent description of the macroscopic ferroelectric devices in terms of device nanostructure and energetic disorder. Apart from being fundamentally relevant, these questions are also practically important, as they provide unconventional guidelines

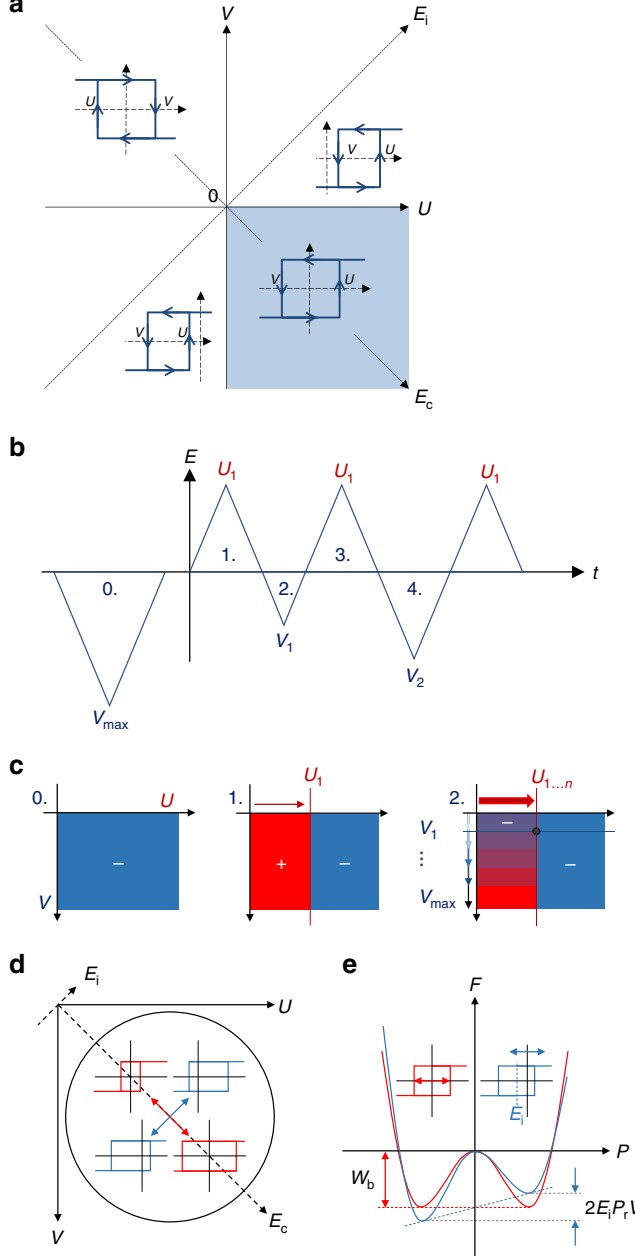

**Fig. 1** Measuring the Preisach distribution. **a** The Preisach plane with both axes systems as explained in the main text. Four different types of hysterons are indicated, the shaded lower right area (4th quadrant) is the probed region of interest. **b** The pulse sequence used to measure the Preisach distribution and **c** the corresponding sweeps in the Preisach plane. Darker shaded areas show the part of the Preisach plane that was switched in the corresponding sweep. **d** Interpretation of the Preisach distribution in quadrant IV of the Preisach plane. Along the diagonal red line ($E_c$) hysteresis curves are symmetric ($E_i = 0$) with variations in the magnitude of $U = -V$. These changes are caused by variation in the energy barrier for switching $W_b$. Broadening along the perpendicular blue diagonal ($E_i$) comes from local interaction fields that shift the hysteresis curves by an offset $\pm E_i$. **e** Corresponding free energy landscape (Landau plot), $P_r$ being the remnant polarization and $V^*$ the critical nucleation site volume

for further development of the ferroelectric materials with superior properties, that can be applied in both the regular single-bit and the emerging multi-bit data storage in single ferroelectric memory elements.

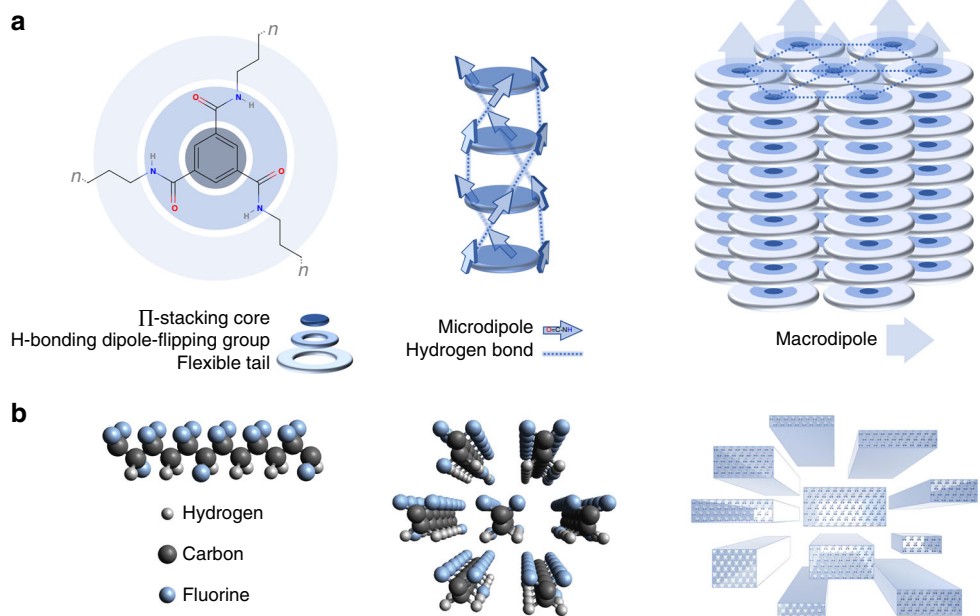

**Fig. 2** Primary, secondary, and tertiary structure of the examined materials. **a** BTA and **b** P(VDF-TrFE)[58]. Discotic BTA molecules stack in a head-to-tail manner forming helical hydrogen-bonded microdipole system. Molecular stacks further self-assemble into a columnar hexagonal lattice. P(VDF-TrFE) copolymer chains, consisting of all-trans conformation $-CH_2-CF_2-$ and $-CHF-CF_2-$ segments, pack pseudo-hexagonally within crystallites. Semi-crystalline nature of the material results in these crystallites being surrounded by an amorphous substance

## Results

**Experimental Preisach distribution.** Thin film capacitor devices of BTA-C10 and P(VDF-TrFE) were fabricated and characterized as described in the Methods section. The PD is measured using the triangular wave sequence in Fig. 1b. With this wave sequence we probe an increasing area of the Preisach plane, as shown in Fig. 1c. Differently to most methods to measure the PD, we do not assume symmetry and probe the whole 2D Preisach plane. More details on this method can be found in the Methods section.

The PD is generally assumed to be a two-dimensional Gaussian function of $U$ and $V$[3,4,6]. However, the model being universal and largely mathematical, fits to it do not necessarily reflect any physical reality of the ferroelectric layers and previously reported attempts to measure the PD experimentally on various ferroelectric materials devices lacked a physical interpretation of the acquired parameters[2,3,6,31]. It has been commonly agreed that domains within a ferroelectric layer experience local fields, different from the macroscopic applied field, that govern the polarization switching processes[32,33]. The Preisach model allows to discern two types of those characteristic fields—a switching (coercive) field $E_c$ and an imprint (interaction) field $E_i$—with a defined distribution in each. It is therefore more appropriate from a physical perspective to change to the variables $E_i = \frac{U+V}{\sqrt{2}}$ and $E_c = \frac{U-V}{\sqrt{2}}$[32], which corresponds to a 45° rotation of the axes shown in Fig. 1a. This way each hysteron has a symmetric hysteresis loop with an intrinsic coercive field $E_c$, centered around the local internal field $E_i$. This gives the normalized distribution

$$N(E_i, E_c) = \frac{1}{4\pi\sigma_i\sigma_c} \exp\left(-\frac{(E_i - m_i)^2}{4\sigma_i^2}\right) \exp\left(-\frac{(E_c - m_c)^2}{4\sigma_c^2}\right),$$

(1)

where $(m_i, m_c)$ is the center of the distribution. The width of the distribution in the directions of both axes is $\sigma_{i/c}$, and is related to

the domain structure in the material and their interactions, as will be discussed in more detail below. The assumption of a Gaussian distribution is validated by the quality of the fits as demonstrated in Supplementary Figure 2, whereas fitting with other distribution functions such as lognormal and Lorentzian fails.

The results of the measurement and fitting procedure described above are shown in Fig. 3 for BTA and P(VDF-TrFE). A nearly circular ($\sigma_i \approx \sigma_c$) Gaussian PD is found for the BTA devices. The center of the distribution virtually lies on the line $U = -V$, corresponding to a distribution of the internal fields $E_i$ with a mean value $m_i = 0$, which is to be expected for materials with non-imprinted macroscopic hysteresis loops. In contrast to the BTA distribution, the distribution for P(VDF-TrFE) has a pronounced ellipsoid shape with $\sigma_i \ll \sigma_c$. The values found here correspond to those found previously by Tsang et al. with a simplified method[16], although our value of $\sigma_i$ is lower (1.3 vs. 4.9 V μm⁻¹), which is likely due to differences in crystallinity between the devices. Deviations from $m_i = 0$ are likely due to non-switching background currents and experimental noise.

There are multiple classical models that are potentially applicable to extract the relevant parameters of the ferroelectric from experimental switching data like KAI[20,34], Merz[35] and hybrid-Merz[36], and Du-Chen[37]. However, most of these are limited to ideal crystals and epitaxial films, only work at higher fields and shorter time-scales or do not include temperature dependence[23]. When considering imperfect polycrystalline ferroelectric thin-films, it is now becoming generally accepted that the ferroelectric switching is nucleation-limited with a wide distribution of switching times[23–25,38]. According to this concept, the rate-determining step for polarization switching is the formation of a critical nucleus with a defined activation energy density $w_b$ and volume $V^*$. This leads to the coercive field value being directly related to this nucleation process.

On basis of the Landau–Devonshire theory and including thermal activation, the coercive field $E_c$ can be approximated

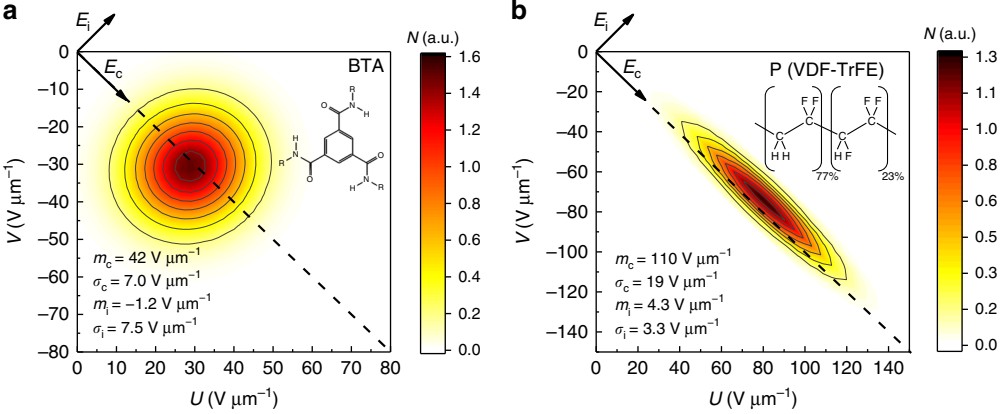

**Fig. 3** Fitted normalized Preisach distribution. **a** BTA at 55 °C and **b** P(VDF-TrFE) at room temperature. The rotated coordinate system is shown, and the dashed line indicates $U = -V$. The four fit parameters are given in each panel

using an equation constructed by Vopsaroiu et al.[39,40]:

$$E_c \cong \frac{w_b}{P_r} - \frac{k_B T \cdot \ln\left(\nu_0 t \cdot (\ln(2))^{-1}\right)}{P_r V^*} \qquad (2)$$

Here, $P_r$ is the remnant polarization, $k_B$ is the Boltzmann constant, $\nu_0$ is an attempt frequency (e.g., a typical frequency of optical phonons), and $t$ is the measurement time, which is a reciprocal frequency of the probing field signal. Within this framework, the coercive field is linearly proportional to $w_b$, the energy barrier per unit volume between the two polarization states, and the temperature $T$. This coercive field expression represents an extrinsic threshold value and is far lower than the theoretical intrinsic coercive field, coming from the pure Landau–Ginzburg theory for an ideal crystal[41], which allows it to correctly describe various ferroelectric systems with different degrees of disorder.

This TA-NLS theory has been successfully applied to analyze the switching processes in inorganic as well as organic ferroelectrics[29,30,42,43]. Importantly, it is compatible with the idea of energetic disorder and can be modified by introducing a distribution in energies and random local fields. Specifically, for a polycrystalline material, it is natural to introduce the disorder by taking a distribution of the individual critical domain sizes $V_k^*$. This leads to a statistical local variation of the total energy barrier $W_{b,k} = w_b V_k^*$. The variations in $W_{b,k}$ will symmetrically distort the free energy landscape by changing the depth of the energy valleys as shown in Fig. 1e. In this scheme to introduce disorder, the characteristic energy density $w_b$ is interpreted as a constant material property. For the specific case of BTA, $w_b$ is related to the activated rotation of permanent amide dipole upon switching. It contains contributions from steric forces and, in particular, Coulombic forces associated with the breaking and reformation of the intermolecular hydrogen bonds that will largely be the same for all molecules in the system. In the case of other polycrystalline ferroelectrics, e.g., perovskites, $w_b$ would correspond to the energy barrier for the ion displacement event.

As illustrated in Fig. 1d, e, local interaction fields within the ferroelectric can skew the free energy landscape, leading to nonzero values of $E_{i,k}$, causing a sideways shift of the hysteron. The total effective energy barrier for switching of a specific hysteron $k$ then becomes $W_{eff,k} = \left(w_b \pm P_r E_{i,k}\right) V_k^*$. The

corresponding coercive field consequently changes to:

$$E_{c,k} \cong \frac{w_b}{P_r} \pm E_{i,k} - \frac{k_B T \cdot \ln\left(\nu_0 t \cdot (\ln(2))^{-1}\right)}{P_r V_k^*} \qquad (3)$$

We can now relate the change in the energy landscape back to a change in the hysteresis loop of a hysteron. Varying $W_{b,k}$ (through $V_k^*$) results in symmetric loops of varying width, following the red $U = |V|$ diagonal in the Preisach Plane in Fig. 1d. Varying $E_{i,k}$ results in hysteresis loops with the center shifted away from $E_i = 0$. In the Preisach plane, this corresponds to moving parallel to the blue diagonal in Fig. 1d.

In inorganic polycrystalline and epitaxial ferroelectric thin films, the distribution in local fields is generally understood to originate from a multitude of different types of defects and other pinning sites[44]. They might be intrinsic, such as 90° domain walls or dislocation dipoles, as well as extrinsic like surface and bulk contaminants. They are the origins of the commonly used empirical concepts of random-bond and random-field (corresponding to $E_c$ and $E_i$, respectively), which influence the local hysteresis loops as discussed above[44,45]. We use the polycrystalline ferroelectric BTA and the semi-crystalline P(VDF-TrFE) with their well-described morphology as a model system to explain the suggested alternative origin for the variation in $E_{c,k}$ and $E_{i,k}$.

Based on the discussion above, one can evaluate the energy disorder from the experimental PD parameters using Eq. (2). For BTA, we find a mean critical domain size $V^*$ of around 8 nm³ (using $w_b = 0.1$ eV nm⁻³) and a total mean activation energy $W_b$ of ≈0.8 eV, which are realistic numbers for these molecular materials[29,46]. The broadening $\sigma_c$ can be explained by a standard deviation in $V^*$ of ≈0.3 nm³. A similar analysis for P(VDF-TrFE) with $w_b = 0.2$ eV nm⁻³ gives a mean $V^*$ of 4 nm³ with ≈0.4 nm³ standard deviation, which is relatively greater than that of the BTA. Characteristic $w_b$ values are taken from fitting data displayed later in the text.

The broadening along the $E_i$ axis rises from different effects. Aligned BTA layers consist of molecular columns of varied length, depending on the degree of disorder (Fig. 2a). The tight columnar-hexagonal packing of BTA (inter-column distance ~2 nm, as obtained from XRD measurements) assures strong interaction between the columns[30], causing each (fragment of a) column to experience the polarization fields of its neighbors. The co-axial organization will make the pair interactions dependent on the sizes of the columns in the pair, with a magnitude that will be of the order of the product of the two total

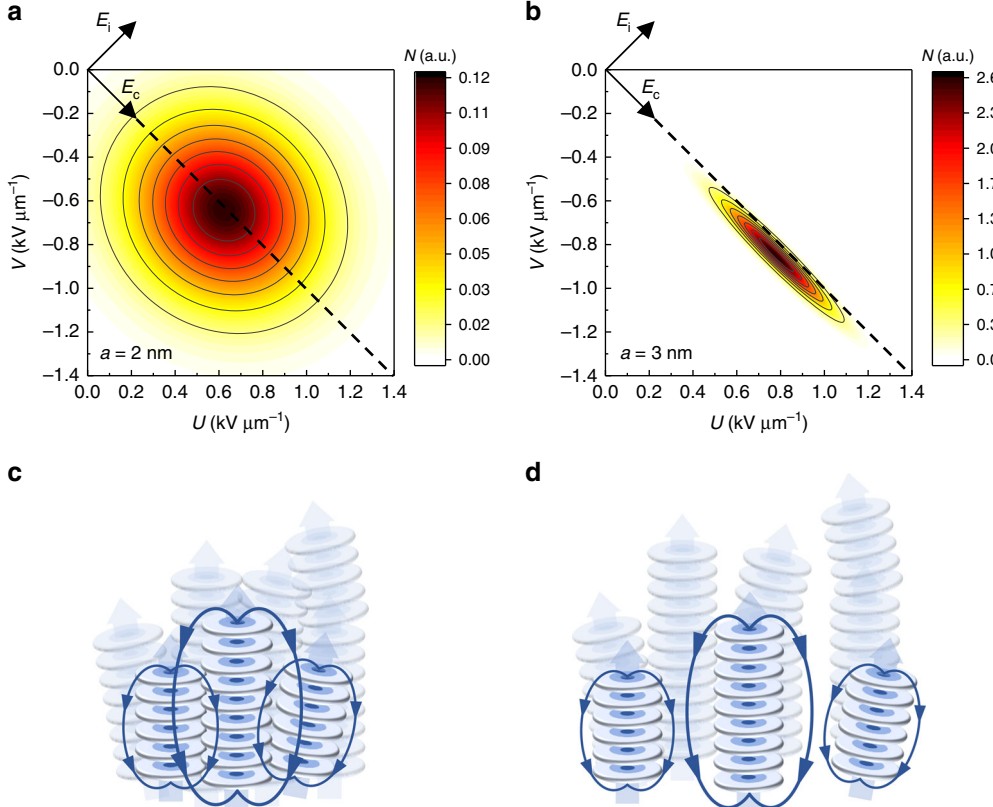

**Fig. 4** Simulated Preisach distributions. The distributions are extracted from the simulated polarization switching for BTA with a columnar separation $a$ of **a** 2 nm, cf. BTA, Fig. 3a and **b** 3 nm, cf. P(VDF-TrFE), Fig. 3b. The rotated coordinate system is shown and the dashed line indicates $U = -V$. **c, d** Illustration of the model system used for the simulations. Discs represent molecules that stack in a head-to-tail manner to form clusters. Thick transparent arrows indicate the macrodipole. The size of the clusters and the electrostatic interactions (blue lines) between them determine the form of the Preisach distribution

dipole moments. Hence, each domain $k$ experiences external fields $E_{i,j,k}$ from its neighboring domains $j$ that will add up to $E_{i,k}$ and, importantly, have a magnitude $\propto P_r V_j^*$. Therefore, strongly interacting systems as BTA are expected to have a broad distribution in $E_i$.

P(VDF-TrFE), in contrast, is a semi-crystalline polymer, possibly reaching >50% crystallinity at the right processing conditions. P(VDF-TrFE) is known to form long rod-like pseudo-hexagonally packed polymer chains (Fig. 2b)[47]. X-ray diffraction measurements show a narrow distribution of the lattice parameters of packed polymer chains, though do not disprove the existence of amorphous matter[48,49]. In fact, the presence of substantial amounts of amorphous matter between crystallites has been recently shown to be crucial for the anomalous piezoelectric behavior of P(VDF-TrFE)[48]. Therefore, the broad $E_c$ distribution in Fig. 3b is, as for BTA, likely to be related to the distribution in the crystallite volume. In contrast to BTA, the narrow $E_i$ distribution would be consistent with weaker interactions between crystallites due to relatively large inter-crystallite distances filled with amorphous material. A similar "effectively independent domains" argument has been used when explaining the stability of the intermediate polarization states in P(VDF-TrFE) capacitors[19].

**Simulated Preisach distribution.** We can validate the above explanation for the distribution in $E_i$ using a simple model. As the argument is based solely on electrostatic interactions, we will reduce the system to a collection of point dipoles and their interactions. The location and orientation of these dipoles is based on the hexagonal columnar structure of BTA, which includes disorder. The dipoles are organized into clusters that correspond to the domain size, see Fig. 4c, d. We calculate the electrostatic interactions between all dipoles within a certain cut-off range, from which we obtain the energy required to flip a cluster. Starting with all clusters polarized in one direction, we increase the external electric field step by step. At each step, the energy of each cluster is recalculated. If it becomes energetically favorable to flip a cluster, we flip the direction of all dipoles in this cluster. By sweeping the whole Preisach plane in the same manner as in the experiment in Fig. 1, we can extract the PD. A more detailed description of the model can be found in Supplementary Note 1.

The resulting PD is shown in Fig. 4 for different cluster spacing $a$. The simulation results fully reproduce the experimental form of the PD and fit well to a bivariate Gaussian function (see Supplementary Figure 4). When the inter-cluster distance is increased from 2 nm (Fig. 4a, c) to 3 nm (Fig. 4b, d), a narrower distribution with a limited spread in $E_i$ is obtained. Increasing the cluster separation further would eventually lead to a virtually 2D PD. This proves that the internal electrostatic interactions are responsible for the broadening along the $E_i$ axis. This applies to ferromagnetic materials as well[50].

The morphology of P(VDF-TrFE), with its weakly interacting crystallites, can be compared to the case of the weakly interacting clusters when a relatively large distance (≥3 nm) is chosen. In contrast, BTA has an intercolumnar distance of around 2 nm, ensuring a strong interaction between columns and resulting in the broadened, circular distribution observed in Fig. 3b. We can

thus directly relate the shape of the PD to the micro- and mesostructure of the materials. Extending the above arguments to inorganic ferroelectrics like PZT or BFO that have a polycrystalline structure containing defects, grain boundaries, etc., but little amorphous (non-polar) phases, one may expect a mostly "BTA-like" circular PD. Indeed, for the few systems for which the PD was actually measured, a circular form was obtained[4,6].

**Switching kinetics**. Polarization switching in polycrystalline/disordered ferroelectrics is mostly dispersive with a broad distribution of characteristic switching times. Several switching time distributions, from exponentially broad[23], to log-Lorentzian[24] and log-normal[25–27], have been introduced into the classical KAI theory to successfully fit the experimental switching current transients. It has been suggested that these distributions rise from variation in local fields, however the argument requires certain assumptions[38]. We previously examined the macroscopic switching kinetics of BTA and found the switching process to be nucleation-limited. A broad log-normal distribution of characteristic switching times within the KAI theory was needed to describe the experimental switching current transients[25]. Above we have shown how structural disorder translates into the energetic disorder that is mapped on the Preisach plane. Next, we apply the TA-NLS theory to connect the microscopic details to the macroscopic switching kinetics. This will elucidate the relation between the PD and the characteristic switching time distribution that result in the observed dispersive switching kinetics.

Being able to address different parts of the PD, we will also measure the experimental switching kinetics of discrete PD sections. For this we use a measurement protocol with the pulse signal sequence as shown in Fig. 5, which gives us log-normal switching current transients for each point in the Preisach plane at the same constant applied field. From these transients, we extract the switching time $t_{sw}$ values and map them onto the Preisach plane. More details on this protocol can be found in the Methods section. The result in Fig. 6a shows a stark increase of the time needed for switching when moving further from the axes.

To better understand the observed trend, one may analyze limiting cases of the measurement: keeping $E_{set,V} = E_{app}$ (scenario A, along the red line in Fig. 6a) or $E_{set,U} = 0$ (scenario B, along the blue line) constant, while gradually changing the other setting field. This corresponds to probing "slices" of the PD—e.g., a particular range of $U$ with the whole available range of $V$. A set of current transients for these two cases is given in Fig. 6c, d. For case A (Fig. 6c), the current peak position (=switching time)

moves towards longer time scales with increasing $E_{set,U}$. The corresponding switching time values are plotted in Fig. 6b as red full circles. To confirm that the used scheme indeed gradually fills the whole PD, the switched polarization $\Delta P$ is simultaneously obtained by integrating the switching current transient. It is plotted as red open symbols and saturates at twice the remnant polarization. The PD projection on the $U$ axis can then be obtained by differentiation of the polarization filling data and is again of Gaussian form (Fig. 6b, dashed grey line).

The growing trend of the switching time can be understood as follows. Increasing the elimination boundary $E_{set,U}$ means that low coercive field hysterons are gradually removed, the result of which is an increased switching time. This implies that by increasing $E_{set,U}$ we prevent the system from accessing the "easiest" nucleation centers, leaving only the ones with higher activation energy, which need more time to be overcome.

We can also set $E_{set,U}$ to zero and probe "horizontal" slices of the PD by changing $E_{set,V}$ (scenario B). Even though increasing the setting field $E_{set,V}$ results in adding more hysterons similarly to scenario A, in this case the switching current peak position (thus the switching time) remains roughly unchanged, as seen in Fig. 6d. This constant switching time (blue full circles in Fig. 6b) sits close to the minimum value obtained from the measurement mode A. This tells that the switching time is at its minimum if hysterons with the smallest coercive field value—good nucleation sites corresponding to $U \approx 0$—are available, no matter which fraction of the PD is switched. These are clear indications for the nucleation limitation of the switching process.

The experimental switching time dependence on the Preisach coordinates of the hysteron can be explained theoretically using the same TA-NLS model (Eq. (2)). Within this formalism, the complete switching time $t_{sw}$ at the applied field $E_{app}$ is expressed as[40]:

$$t_{sw} \cong \frac{1}{\nu_0} \cdot \exp\left(\frac{\left(w_b - P_r E_{app}\right) V^*}{k_B T}\right). \qquad (4)$$

It is directly related to the rate of reversal of the critical characteristic nucleation site and does not account for the domain growth as this is assumed to be negligible in comparison. Similar to the analysis above, we introduce energetic disorder in this equation via a distribution in the critical domain size $V^*$, which we further express in Preisach parameters using Eq. (3). Substitution in Eq. (4) leads to Eq. (5) that gives the switching

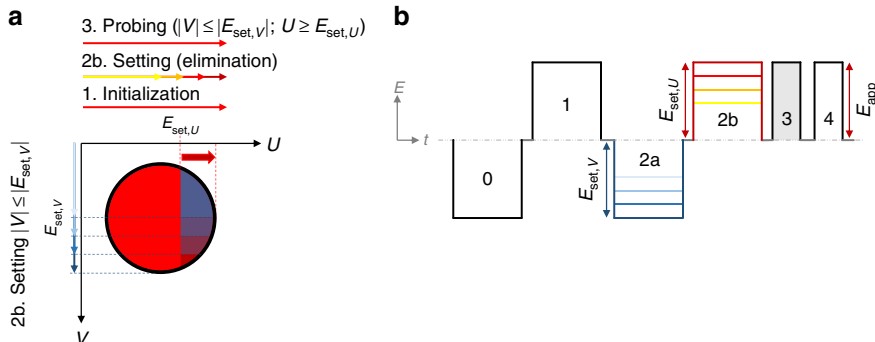

**Fig. 5** Probing the switching kinetics of different parts of the Preisach distribution. **a** Schematics of the measurement in the Preisach plane. **b** The corresponding pulse sequence. Red indicates positive and blue negative poling. The red thick arrow indicates the measured PD projection on the $U$ axis. Steps 0 and 1 reset the polarization state, the setting field $E_{set,V}$ and $E_{set,U}$ in step 2 are gradually changed to probe the whole Preisach distribution. The switching kinetics is measured in step 3 at the applied field $E_{app}$ (amplitude unchanged), while step 4 is used for non-switching background current evaluation. The pulse duration of all steps is kept constant

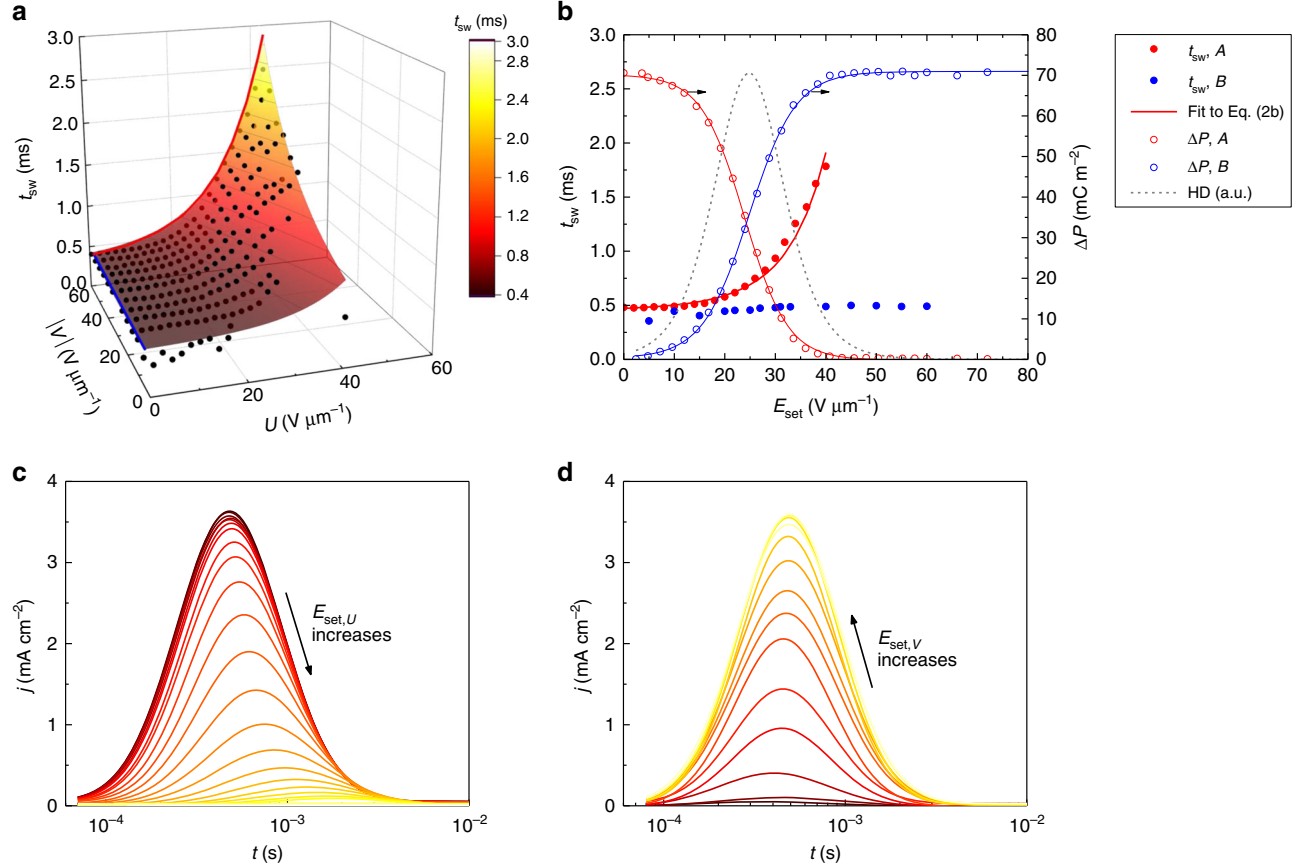

**Fig. 6** Switching kinetics of different parts of the Preisach distribution. **a** Switching time dependence on the Preisach coordinates of a BTA device (measured at a constant applied field). Experimental values (black circles) were obtained using the measurement procedure from Fig. 5 at $T = 55\,°C$ and fitted to bivariate Eq. (5) (see Supplementary Table 2 for fit parameters). **b** Switching time $t_{sw,k}$ (full circles) and switched polarization (open circles) plotted vs. the setting field $E_{set}$ for $A$ (red) and $B$ (blue) measurement modes. The $t_{sw}$ values have been chosen as the time when the experimental switching current transients—panel **c** for case $A$ and panel **d** for case $B$—reach their maximum

time of the domain $k$ with the Preisach coordinates $E_{c,k}$ and $E_{i,k}$:

$$t_{sw,k} \cong \frac{1}{\nu_0} \cdot \exp\left(\frac{\left(w_b - P_r E_{app}\right) \cdot \ln\left(\nu_0 t \cdot (\ln(2))^{-1}\right)}{w_b - P_r\left(E_{c,k} \pm E_{i,k}\right)}\right). \quad (5)$$

When transformed to bivariate, this formula successfully describes the experimental exponential switching times as shown by the colored surface in Fig. 6a, proving the validity of the TA-NLS model. The nucleation barrier value used for fitting was $w_b = 0.1\,\text{eV nm}^{-3}$, which is well in the $0.002$–$0.25\,\text{eV nm}^{-3}$ range that has been reported for other materials[42,43]. We also needed to add a "floor" switching time $t_0 \approx 0.4\,\text{ms}$ to Eq. (5), which is a minimum switching time of the fully filled PD, and is a natural observation[51]. A variable change $E_i = \frac{U+V}{\sqrt{2}}$ and $E_c = \frac{U-V}{\sqrt{2}}$ was used for plotting and projection fitting. This makes $E_{set}$ equal to projections of the effective coercive field ($E_c \pm E_i$) on the $U$ and $V$ axes: $(E_c \pm E_i) = \sqrt{2}U = \sqrt{2}V$. We remark that, in principle, it is sufficient to experimentally obtain only the $U$ and $V$ projections of the data of Fig. 6a to acquire the fitting parameters in Eq. (5), see Supplementary Figure 6. This would reduce the experimental costs significantly, as compared to obtaining and analyzing the whole 2D switching time dependence.

By convoluting the switching time $t_{sw}$ dependence on the Preisach parameters (Eq. (5), Fig. 6a) with the actual PD (Eq. (1),

Fig. 3), we are now able to obtain the characteristic switching time distribution $F(t_{sw})$ of the macroscopic device, see Fig. 7a, black line. The resulting distribution is close to log-normal, with small discrepancies at short times due to limitations of the experimental switching time evaluation, c.f. dashed red line. This distribution can be incorporated in the equation for the temporal dependence of polarization reversal from the KAI-based NLS theory (Eq. (6))[23,24]:

$$\Delta P(t) = 2P_r \int_{-\infty}^{+\infty} \left[1 - \exp\left(-\left(\frac{t}{t_{sw}}\right)^d\right)\right] F(t_{sw})\,dt_{sw} \quad (6)$$

The time derivative of this equation gives the full switching kinetics $J(t) \propto d\Delta P/dt$. This methodology leads to a very good fit to the macroscopic experimental switching current transients, see Fig. 7b. By this we show that the PD, the switching time distribution and the macroscopic polarization switching kinetics are directly connected—the fits to the experimental data in Figs. 3, 6, and 7 use a single, consistent parameter set (see Supplementary Table 2)—and result from the nanoscale morphology as shown in Fig. 4.

Having shown to be able to successfully perform this type of characterization on BTA devices, the same methodology was used on P(VDF-TrFE) devices. In general, the switching process of P(VDF-TrFE) thin-films has been the topic of on-going

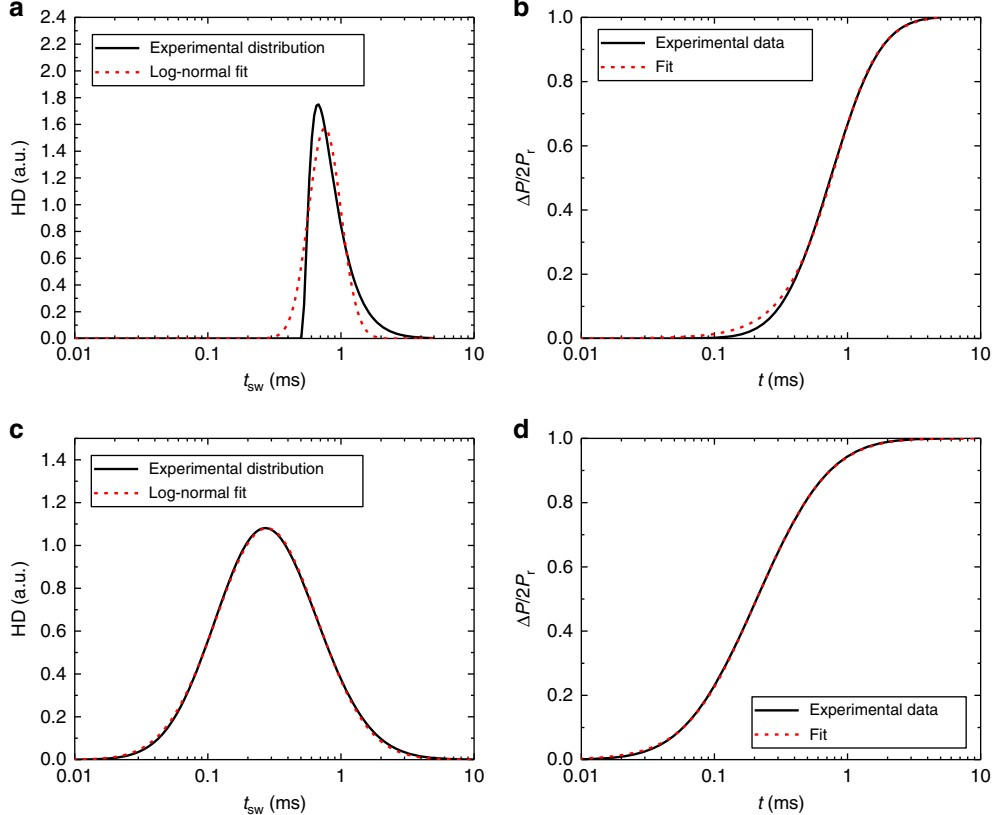

**Fig. 7** Extracted total switching time distributions. The distributions are obtained from the convolution of experimentally obtained switching time dependence on Preisach coordinates and corresponding experimental Preisach distribution for **a** BTA and **c** P(VDF-TrFE), and a lognormal distribution fitted to that (red dashed line). The resulting distributions are used to fit the experimental macroscopic switching kinetics that are plotted in an integrated form in panel **b** for BTA and panel **d** for P(VDF-TrFE). The fit parameters can be found in Supplementary Table 2

discussions due to multitude of different experimental results and matching theories[38,52–54], leading to the conclusion that experimental details have a major influence on the device behavior. We find that, similarly to BTAs, the experimental switching current transients of our P(VDF-TrFE) devices obtained by the same measurement technique (Fig. 5), are of log-normal form (see Supplementary Figure 7) and match the results from regular switching kinetics measurements that were found before[27,53]. The characteristic switching time dependence on the setting field (Supplementary Figure 7c) in boundary case $A$ shows an exponential dependence and can be successfully fitted using Eq. (5) with $w_b = 0.2$ eV nm$^{-3}$, which is twice higher than for BTA due to higher rotational energy barrier of the $-CH_2-CF_2-$ chains, compared to the H-bonded O···H–N network of BTAs[55]. The switching time values in case $B$ reach a plateau value $t_0$ of around 0.2 ms, which does not exceed the minimum switching time of case $A$. This implies that the switching process in our P(VDF-TrFE) devices at these experimental conditions is also nucleation-limited. Following the same analysis path as for BTA, we found that also P(VDF-TrFE) is characterized by a log-normal switching time distribution (see Fig. 7c), which is broader than BTA's due to relatively wider distribution in the activation volume as shown earlier in the text. A very good fit of the experimental switching kinetics was obtained by evaluating Eq. (6) with this experimental switching time distribution, see Fig. 7d.

**Intermediate states.** Ferroelectric materials with a high level of energetic disorder have an interesting application potential, which arises from the possibility to gradually fill their broad PD. By

using such intermediate polarization states, it is possible to construct multibit memories. For example, a 3-bit memory with eight polarization states has been demonstrated for inorganic BFO and PZT[56]. Whereas these examples suffer from non-binary readout signals, Khikhlovskyi et al.[18] demonstrated a 3-bit memory on P(VDF-TrFE) with binary readout, that is based on the validity of the Dipole Switching Theory (DST). The DST, in turn, assumes the PD to be one-dimensional on the line $U = -V$[8,9,18].

The mechanism behind this 3-bit memory is shown in Fig. 8. Using, e.g., the pulse sequence 123 in Fig. 8a, a "101" state is created with the corresponding PD in Fig. 8b. To construct any state between "000" and "111" one only needs to change the signs of pulses 1, 2, and 3. The readout current during pulse 4 gives a distinct peak or valley for each of the 3 bits, as sketched in Fig. 8c.

It has been recently demonstrated by Katsouras et al. that these intermediate states are of remarkable stability, in contrast to the conventional theory which assumes the local energy minima to be fundamentally thermodynamically unstable[19]. This counterintuitive observation can now be rationalized by the electrostatic decoupling of the ferroelectric domains in P(VDF-TrFE) due to its semi-crystalline structure.

As explained, the construction of the multi-bit data storage scheme introduced in ref. [18] is enabled by the narrow elliptical PD that seems to be specific for P(VDF-TrFE) owing to the decoupled crystalline domains with a significant size distribution. For strongly interacting domains which give circular distributions, as in BTA or inorganic ferroelectrics, this strategy will not work as the readout of the different bits would overlap, see Fig. 8d, e. To create a multibit memory in these materials one therefore must resort to the non-binary scheme shown in

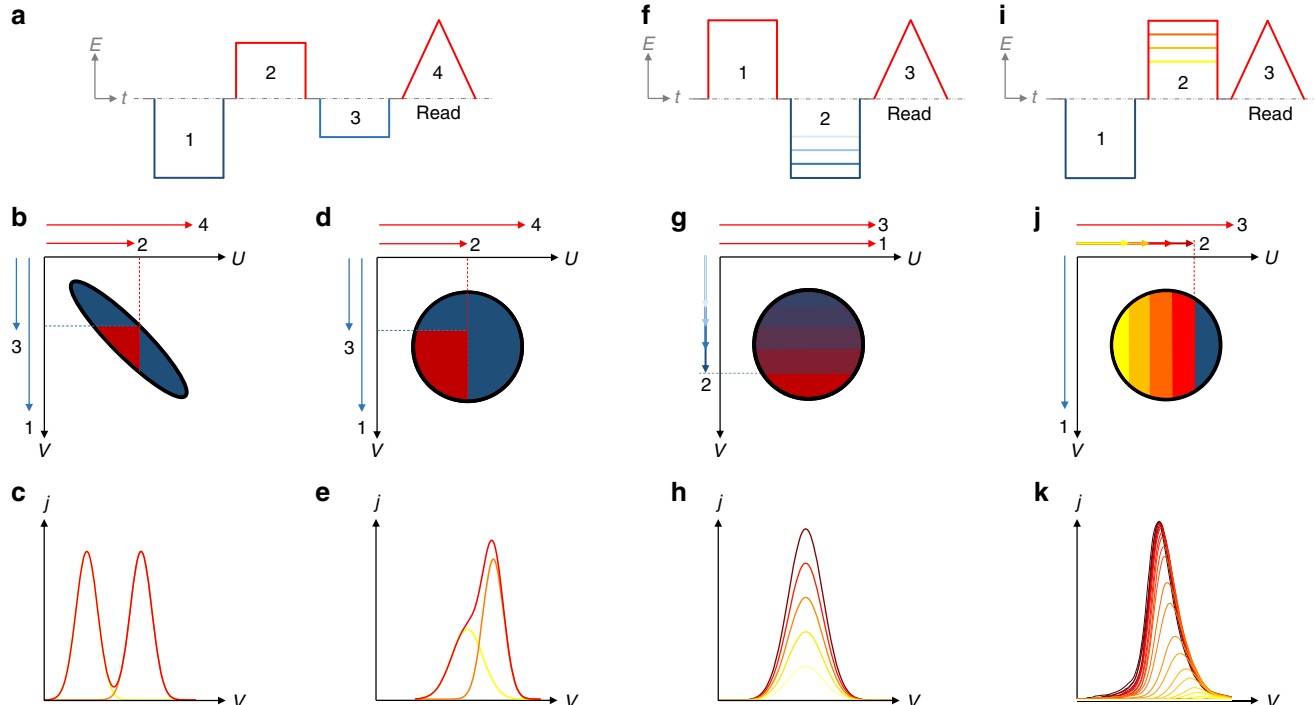

**Fig. 8** Multi-bit memory writing and reading. **a** shows the pulse sequence (123) for setting the memory, with a subsequent sawtooth pulse (4) to read it out. The Preisach distributions after writing are shown in **b**, **d**, and the current during the readout pulse in **c**, **e**. Results are shown for the elongated PD, e.g., P(VDF-TrFE), on the left and the circular PD, e.g., BTA, on the right. Due to the round form of the PD in panels **d**, **e**, the two read-out peaks are indistinguishable. A different multi-bit data storage protocol is needed, see panels **f–k**. Two implementation modes are possible—down-filling (**f–h**) and up-filling (**i–k**)—and would give similar results in terms of switched polarization, see Supplementary Figure 8. The latter results in a shift of the peak position during the read-out

Fig. 8f–k. In this scheme, the PD is increasingly filled based on the amplitude of writing pulse 2. Integrated polarization results for this can be found in Supplementary Figure 8a.

## Discussion

The examined BTA and P(VDF-TrFE) can be described within the same theoretical framework as we outlined above, even though they are very different systems in terms of energetics and morphology. While a wide distribution in domain sizes is common for both, the key distinction between the two systems lies in the inter-domain coupling, Fig. 3, with the strong (weak) coupling for BTA (P(VDF-TrFE)) giving rise to a circular (elongated) PD. As demonstrated in Figs. 6, 7, the same factors that explain the broadening of the hysteresis loops can also quantitatively explain the experimental switching kinetics both at microscopic (for hysterons on different parts of the Preisach plane) and macroscopic conditions. These insights generally apply to any ferroelectric material of polycrystalline and semi-crystalline structure, including inorganic ones. This proves that the PD of hysterons is not only a mathematical construct that can empirically fit polarization curves, but also reflects a physical reality of interacting domains of variable size.

From a practical perspective, these findings provide guidelines how ferroelectric materials can be disorder-engineered depending on their application field. Differently to the universally accepted idea that obtaining ideal single-crystals is the key to excellent ferroelectric properties, the insights of the paper suggest an alternative, potentially easier-implementable approach. The "ideal" ferroelectric behavior, i.e., with rectangular polarization loops and immediate polarization switching, would be achieved in systems consisting of ferroelectric particles, like crystallites or

nanoparticles, with a narrow size dispersion (ideally, mono-disperse). The particles should be electrostatically isolated from each other, for example by embedding them in a dielectric matrix. The optimal isolation distance would depend on the dielectric properties of the two phases. In this way, due to the limited variation in $E_c$ and $E_i$, close-to-ideal $P-E$ hysteresis curves and fast non-dispersive polarization switching would be expected. Additionally, the macroscopic coercive field would be lower compared to the single-crystals for which the coercive fields approach the intrinsic value. This would be accompanied by a highly stabilized microscopic as well as macroscopic polarization due to reduced interdomain interactions. Technologically, such conditions could be implemented in high-quality semi-crystalline or (nano)composite ferroelectric materials. Such materials with a quasi-1D PD are required for conventional single-bit memories.

To facilitate the construction of multi-bit memory devices, different design rules are proposed. For memory devices with binary data readout, ferroelectric materials with a strongly elongated cigar-shaped (ideally, two-dimensional) PD are to be sought after. Polydisperse materials, having weakly-coupled domains of wide size distribution (i.e., $\sigma_i \ll \sigma_c$), would be preferred for this application. The broad symmetric (circular, 3D) PD (i.e., materials with polydisperse, strongly interacting domains) serves for multi-bit data storage with non-binary readout. Again, the form of the PD could be tuned by engineering of the nanostructure of the ferroelectric material. Hence, the presented results propose design rules for materials with controlled disorder for both conventional single-bit and prospective multi-bit ferroelectric memory applications.

Summarizing, we have uncovered a physical basis for the otherwise mathematical Preisach model in organic ferroelectrics. We have shown how the shape of the PD can be directly related

to, and even derived from the morphology and structural disorder in the material, by the example of two archetypal organic ferroelectrics—polycrystalline BTA with strong inter-domain coupling and semi-crystalline P(VDF-TrFE) with weak inter-domain coupling. The broadness of the PD has been directly related to a distribution in domain sizes and the electrostatic interactions between these domains. In contrast to the previous theoretical and phenomenological approaches, the model provides a formal physical explanation for the dispersive switching kinetics observed in most ferroelectrics and the underlying distribution in switching times. By measuring the switching kinetics of discrete parts of the Preisach plane, we can directly extract this distribution. As a result, a full and consistent description of both the hysteresis loops and the macroscopic switching kinetics in terms of device morphology and energetic disorder is obtained. These findings have direct implications for the design of controlled-disorder ferroelectric materials with 1D, 2D, and 3D PDs, that serve conventional as well as multi-bit ferroelectric memory applications.

## Methods

**Materials and device fabrication**. BTA-C10 (tridecylbenzene-1,3,5-tricarboxamide) material was synthesized using a standard method as described in the previous work[57]. P(VDF-TrFE) (77%/23%) was purchased from Solvay and used as received. Thin-film spin-coated metal-ferroelectric-metal capacitor devices were fabricated as described previously (see Supplementary methods)[18,25]. Saturated $P-E$ curves were measured for the different compounds and can be found in Supplementary Figure 1. A remnant polarization of $P_r \approx 35$ mC m$^{-2}$ and $P_r \approx 65$ mC m$^{-2}$ was found for BTA-C10 and P(VDF-TrFE), respectively, which matches the previously reported values of average devices for both materials[19,30].

**Measurement of the Preisach distribution**. The PD is measured using the triangular wave sequence shown in Fig. 1b. First, the sample is fully poled in one direction, negative in this case, to erase any previous polarization states. Subsequently, the voltage is alternated with a fixed ramping speed, keeping the positive turning point voltage constant while increasing the negative turning point voltage. This way, an increasing area of the Preisach plane will be switched during each sweep, as shown in Fig. 1c. This measurement is repeated for increasing positive turning point voltages to sweep the whole Preisach plane. Essentially, this implementation is similar to the commonly used measurement to obtain first order reversal curves (FORCs), though to achieve full coverage of the Preisach plane the maximum probing voltages are gradually changed. We only probe the fourth ($U >$ 0, $V < 0$) quadrant, which is the relevant quadrant for proper ferroelectric materials like BTA and P(VDF-TrFE). No or few hysterons are found away from this part in non-imprinted materials[2,3].

We found this technique to be more accurate compared to other reported attempts to experimentally obtain the PD or the simplified FORC distribution, where a symmetric distribution is constructed from a single FORC measurement with one maximum amplitude[2,3]. Instead, we do not assume symmetry and probe the whole 2D Preisach plane by applying multiple FORC measurements of different amplitudes and step-by-step address the whole Preisach plane[6].

The above measurement procedure typically results in a set of $20 \times 20$ current transients. The obtained current responses are integrated to obtain the switched polarization for each part of the Preisach plane at 400 different ($U,V$) coordinates. The real PD can now be obtained by differentiating this cumulative distribution. Since this would amplify the measurement noise, we instead fit an integrated model distribution function and take the derivative of the fit to obtain the PD.

**Switching kinetics of parts of the PD**. We measure the experimental switching kinetics of discrete PD sections using a measurement protocol with the pulse signal sequence as shown in Fig. 5. By applying square voltage pulses with different amplitude, we firstly reset the device state to fully poled PD (steps 0, 1) and then we prepare a PD part of interest (step 2a, 2b) by setting it opposite to the probing pulse polarity. At step 3, we probe the switching kinetics at a constant applied field (the same for each probed part). This measurement method allows to access the PD parts defined by their projections on the $U$ and $V$ axes. Namely, hysterons with Preisach coordinates in range ($U \geq E_{set,U}$; $|V| \leq |E_{set,V}|$) are probed. $E_{set,U}$ and $E_{set,V}$ values are gradually changed to acquire data from the whole PD. In terms of polarization, the results are cumulative. Step 4 is needed for the estimation of the non-switching background current (mainly due to the RC-time of the system), which is subtracted from the switching current transients obtained at step 3. The applied field amplitude $E_{app}$ for steps 1, 3, and 4 is chosen as a minimum field for reaching full polarization saturation and is kept constant during the whole measurement sequence. A constant probing step length of 50 ms is selected to allow full switching of the device. We also make sure that the up- and down-switching is

symmetric, and no depolarization is present on the time-scale of the measurement, which could be relevant at high temperatures[29].

The measured current transients and more details on the implementation of the method can be found in Supplementary Figure 5. These current transients have a well-defined log-normal shape, as we observed before[25]. As we are interested in the switching time of each measured segment, we will use a characteristic switching time $t_{sw}$, which for simplicity is taken as the time at which the transient switching current reaches its maximum. Based on the NLS model, we assume that switching of the probed PD part is activated by the nucleation agents—hysterons with the lowest possible coercive field—sitting very close to the setting field boundary. The $U,V$ coordinates for every $t_{sw}$ point therefore can be set as ($U = E_{set,U}$, $V = E_{set,V}$) and the values can be mapped on the Preisach plane. The resulting switching times are plotted on the 2D Preisach plane in Fig. 6a.

## Data availability
The data and code that support the findings of this study are available from the corresponding author upon reasonable request.

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

## Acknowledgements

We would like to thank Dr. Andrey Gorbunov for discussions in the early stage of the research. I.U. acknowledges funding by Vetenskapsrådet. T.D.C. acknowledges financial support from the Swedish Government Strategic Research Area in Materials Science on Functional Materials at Linköping University (Faculty Grant SFO Mat LiU No. 2009 00971).

## Author contributions

I.U. and T.D.C. contributed equally to the production of the article. I.U. and T.D.C. performed the experiments, analyzed the data, and prepared the manuscript. X.M. was responsible for the synthesis and characterization of the BTA material. M.K. and R.P.S. supervised the work. I.U., T.D.C., and M.K. wrote the manuscript, all authors provided feedback on the manuscript.

## Additional information

**Competing interests:** The authors declare no competing interests.

