## [Peer Review File · Nature Communications]

Reviewers' Comments:

Reviewer #1:

Remarks to the Author:

This is an interesting paper. The authors employed experimental Preisach distribution analysis to quantify the energetic disorder and elucidate the concomitant dispersive polarization switching kinetics for organic ferroelectrics. The results of the study give a new idea to recognize the physical reality of the polarization switching processes through Preisach model in organic/molecular ferroelectrics. This work is of innovative and the well-organized. In view of the fast developed molecular ferroelectric field, the following literature should be cited (DOI: 10.1021/jacs.8b04600; DOI: 10.1021/jacs.7b07715). This article could be published after clarifying the following questions.

1. For the electrical characterization of BTA-C10, which form are used, polycrystalline or single crystal? The authors are suggested to comment on the measured polarization value with the other reported value of BTA-C10 single crystal.
2. The XRD data should be presented for BTA-C10 and P(VDF-TrFE) film.

Reviewer #2:

Remarks to the Author:

In this paper the authors investigate the validity of the Preisach model in the case of two organic ferroelectric materials. The analysis links the shape of the Preisach distribution to the nanostructure of the materials. The macroscopic switching kinetics experimentally obtained are reproduced by combining the microscopic switching kinetics and the Preisach distribution. The analysis is well developed and the technical quality is good. In contrast, the novelty of the work is not clear and its implications and relevance are, in my opinion, not adequately explained.

More in detail, the introduction provides an overview on the use of the Preisach model for ferroelectrics and on the models currently used to describe the switching kinetics in ferroelectrics. The authors claim that the current models are complex and require several assumptions. Therefore, after reading the introduction, I understand that the relevance of this work is the study of the switching kinetics by applying of the Preisach model. Unfortunately, it is not clear what are the relevant implications of this understanding. The authors mentioned that the relevance of this work is the fundamental understanding of the physics behind and it can find application in the context of multi-bit data storage in single ferroelectric memory elements.

Although I fully agree that the multi-bit data storage is an important field of application, I would advice the authors to revise the manuscript in order to put the work in a more general form highlighting the novelty and providing a clear evidence of the substantial advancements given by the study. Without this important revision I feel that the paper is not of broad interest.

Reviewer #1 (Remarks to the Author):

1. This is an interesting paper. The authors employed experimental Preisach distribution analysis to quantify the energetic disorder and elucidate the concomitant dispersive polarization switching kinetics for organic ferroelectrics. The results of the study give a new idea to recognize the physical reality of the polarization switching processes through Preisach model in organic/molecular ferroelectrics. This work is of innovative and the well-organized.
 - a. We thank the Reviewer for positive evaluation of our manuscript and for recognizing the novelty of the presented insights.
2. In view of the fast developed molecular ferroelectric field, the following literature should be cited (DOI: 10.1021/jacs.8b04600; DOI: 10.1021/jacs.7b07715).
 - a. We value the suggestion of these references, we have included the review and two related papers by other authors in our literature list as new references 13–15.
3. This article could be published after clarifying the following questions. For the electrical characterization of BTA-C10, which form are used, polycrystalline or single crystal?
 - a. The main conclusions of the article arise from the comparison of polycrystalline and semi-crystalline structure materials, correspondingly, BTA-C10 and P(VDF-TrFE). Therefore, BTA-C10 is used exclusively in the polycrystalline form. This is mentioned in the text, e.g. on p. 3 paragraph 4, p. 6 paragraph 5, or conclusions.
4. The authors are suggested to comment on the measured polarization value with the other reported value of BTA-C10 single crystal.
 - a. The remnant polarization of the BTA-C10 and P(VDF-TrFE) devices used in the study match the values reported before for average material devices. An explanatory text (underlined) with references has been added in the Methodology section, Materials and Device Fabrication subsection: “A remnant polarization of $P_r \approx 35 \text{ mC/m}^2$ and $P_r \approx 65 \text{ mC/m}^2$ was found for BTA-C10 and P(VDF-TrFE), respectively, which matches the previously reported values of average devices for both materials, Ref. 16,27”. For the purpose of this article, average and likely more disordered devices were preferred, therefore, technological optimization of the samples for record parameters was not pursued.
5. The XRD data should be presented for BTA-C10 and P(VDF-TrFE) film.
 - a. The used devices are completely standard, see previous point, and the XRD data for such BTA-C10 and P(VDF-TrFE) samples has been reported before. To avoid repetition we choose not to add the XRD diffractograms in this article, and we hope for the reviewer’s understanding. The references to the literature have been present in the main-text. For P(VDF-TrFE) in page 7, third paragraph, “X-ray diffraction measurements show a narrow distribution of the lattice parameters of packed polymer chains, though do not disprove the existence of amorphous matter, **Ref. 46,47**”. For BTA in page 7, second paragraph, the reference to the XRD data (underlined) has been made explicit as “The tight columnar-hexagonal packing of BTA (inter-column distance $\sim 2 \text{ nm}$, as obtained from XRD measurements) assures strong interaction between

the columns”.

Below we reply to the questions and comments of Reviewer #2. Our response is marked in green, the original comments are subdivided and numbered for clarity. All comments have led to additions/changes in the manuscript and/or SI. A separate document with changes highlighted is submitted as material for review only.

Reviewer #2 (Remarks to the Author):

1. In this paper the authors investigate the validity of the Preisach model in the case of two organic ferroelectric materials. The analysis links the shape of the Preisach distribution to the nanostructure of the materials. The macroscopic switching kinetics experimentally obtained are reproduced by combining the microscopic switching kinetics and the Preisach distribution. The analysis is well developed and the technical quality is good. In contrast, the novelty of the work is not clear and its implications and relevance are, in my opinion, not adequately explained.
 - a. We appreciate the Reviewer's remarks about the quality of the work. We hope that the expressed concerns about the novelty and the impact of the findings have been solved in this updated version of the manuscript.
We would like to add that novelty is generally assumed to be implicit and that we therefore are hesitant to include explicit statements about novelty and try to avoid terms like 'new' and 'novel'. We hope the revised manuscript strikes a good balance.
2. More in detail, the introduction provides an overview on the use of the Preisach model for ferroelectrics and on the models currently used to describe the switching kinetics in ferroelectrics. The authors claim that the current models are complex and require several assumptions. Therefore, after reading the introduction, I understand that the relevance of this work is the study of the switching kinetics by applying of the Preisach model. Unfortunately, it is not clear what are the relevant implications of this understanding. The authors mentioned that the relevance of this work is the fundamental understanding of the physics behind and it can find application in the context of multi-bit data storage in single ferroelectric memory elements. Although I fully agree that the multi-bit data storage is an important field of application, I would advice the authors to revise the manuscript in order to put the work in a more general form highlighting the novelty and providing a clear evidence of the substantial advancements given by the study. Without this important revision I feel that the paper is not of broad interest.
 - a. We thank the Reviewer for pointing out the lack of focus on the novelty and implications in the text. The comments have led to an improved and more explicit representation of the main ideas and conclusions of the study, including a new discussion section. To clarify, while the Preisach distribution (PD) relation to the macroscopic switching kinetics is indeed one of our main findings, there are several other points of relevance:
 - i. From a purely technical perspective, an optimized protocol for the measurement of the experimental Preisach distribution and a new methodology for probing the switching time of its intermediate states have been introduced.
 - ii. We show that the form of the PD has a direct relation to the materials' nanostructure, and one can be derived from the other. This proves that

the Preisach model is not merely a mathematical construct but relates to physical device properties. The parameters E_c , σ_c result from the domain size distribution, while E_i , σ_i quantify the interdomain coupling. As a result, we can now clarify the previously unexplained elongated Preisach distribution of P(VDF-TrFE) and the counterintuitive stability of its intermediate polarization states.

- iii. The same parameters explain the switching kinetics, since, as demonstrated, the distribution in switching times originates from the same PD. In this way, the Preisach model now provides a formal physical description (steady-state and kinetics) of a realistic FE material.
 - iv. On the basis of our findings, new and better ferroelectric materials can be engineered, tailored to their intended application: for single-bit memories materials with a quasi-1D PD are required (ideally, with monodisperse decoupled domains); for multi-bit memories with binary readout, materials with quasi-2D PD (polydisperse, decoupled domains) are recommended, while for non-binary readout 3D PD (polydisperse, strongly interacting domains) are preferred. This recommendation for composite materials might be considered unorthodox in the light of the current trends in the field which moves towards single-crystalline ferroelectrics.
- b. Regarding the novelty of the presented results in the context of published literature, the Preisach model, as a general formalism describing disordered hysteretic systems, has of course found widespread application in mathematical modelling of hysteresis loops of (organic) ferroelectric materials. As such, ‘anything’ that puts a (previously lacking) physical foundation under the Preisach model is of broad relevance. Indeed, as detailed above, our approach goes far beyond the classical phenomenological way by relating the PD to the actual physical properties of the ferroelectric layers. This allows for unprecedented characterization, prediction and tailoring of the macroscopic properties based on the microscopic structure of the ferroelectric, and vice versa. We have, amongst others, added a separate discussion section and re-written the abstract and conclusion sections to emphasize these findings. We expect the implemented changes to enhance both academic and practical interest.

Reviewers' Comments:

Reviewer #1:

Remarks to the Author:

The present manuscript, entitled "Physical reality of the Preisach model for organic ferroelectrics", has been thoroughly revised according the reviewer's comments. The questions I mentioned have been properly addressed. Thus, the manuscript can be accepted for publication.

Reviewer #2:

Remarks to the Author:

I feel that the points raised in the previous round of review have been satisfactorily addressed.